# Distributions of Invasive Arthropods across Heterogeneous Urban Landscapes in Southern California: Aridity as a Key Component of Ecological Resistance

**DOI:** 10.3390/insects10010029

**Published:** 2019-01-15

**Authors:** Weston J. Staubus, Savanah Bird, Savannah Meadors, Wallace M. Meyer

**Affiliations:** Department of Biology, Pomona College, Seaver Biology, 175 W Sixth Street, Claremont, CA 91711, USA; staubuw@wwu.edu (W.J.S.); srb42014@MyMail.pomona.edu (S.B.); srm12014@MyMail.Pomona.edu (S.M.)

**Keywords:** Argentine ant, arthropod, insect, invasion, isopod, myriapod, non-native, spider, suburban, urban

## Abstract

Urban systems often support large numbers of non-native species, but due to the heterogeneity of urban landscapes, species are not evenly distributed. Understanding the drivers of ecological resistance in urban landscapes may help to identify habitats that are most resistant to invasion, and inform efforts to model and conserve native biodiversity. We used pitfall traps to survey non-native ground-dwelling arthropods in three adjacent, low-elevation habitat types in southern California: California sage scrub, non-native grassland, and suburban development. We found that non-native species were fewer and less widely distributed in the sage scrub and grassland habitats. Due to the proximity of our sites, differences in propagule pressure is an unlikely explanation. Instead, we suggest that the absence of water subsidies in the sage scrub and grassland habitats increases those habitats’ resistance to arthropod invasions. Comparisons to studies conducted at fragments closer to the coast provide further support for the relationship between aridity and invasibility in southern California. Our findings highlight that inland fragments are important for conserving native arthropod diversity, that models of non-native species distributions in arid and semi-arid urban systems should include aridity measures, and that reducing resource subsidies across the region is critical to mitigating spread of non-natives.

## 1. Introduction

Urban systems are among the most invaded worldwide [1,2]. However, urban landscapes can be highly heterogeneous across short distances [3], as can the distribution of species within them [4,5,6]. Consequently, understanding coarse-scale patterns of invasion (e.g., what species are present in a metropolitan area) may not be informative to conservation practitioners working within an urban setting. 

Models of invasive species distribution often consider two primary factors: propagule pressure and ecological resistance [2,7]. Theoretical predictions and empirical data strongly suggest that greater propagule pressure increases the number of successful invasions [8,9,10]. Conversely, high ecological resistance, the cumulative influence of ecosystem properties and processes that adversely affect the establishment, growth, and spread of introduced species [11] may reduce invasion success, even at the patch scale [12]. While propagule pressure is likely to be high throughout most urban systems due to anthropogenic dispersal, ecosystem characteristics may vary substantially, causing some areas to be much more resistant to invasion [13].

Southern California, one of the world’s largest and most populous metropolitan areas, is interspersed with a multitude of undeveloped ‘natural area’ fragments [13,14,15,16]. At low elevations, these fragments primarily consist of native California sage scrub and non-native grasslands, both of which are well adapted to the region’s arid climate and harbor significant portions of regional biodiversity [6,14,17,18]. In contrast, most urban outdoor spaces, including yards, parks, and gardens, are irrigated to maintain aesthetically pleasing assemblages of non-native plants. Since high environmental stress and low resource availability are often correlated with lower invasibility [19,20], aridity may be a key factor influencing ecological resistance in this and other arid and semi-arid systems. 

To examine ecological resistance in the heterogeneous southern California urban landscape, we surveyed non-native ground-dwelling arthropods in three common low-elevation habitat types: native California sage scrub, non-native grassland, and suburban development. The patches we sampled were immediately adjacent to each other, minimizing potential differences in climate, dispersal, and propagule pressure. In addition, we compared our results to studies of ground-dwelling arthropods conducted in more coastal habitat fragments in an effort to understand how ecological resistance may differ between coastal fragments and more arid, inland fragments. Understanding the mechanisms behind ecological resistance can provide insights into how to mitigate impacts of non-native species in these semi-arid suburban/urban landscapes. 

## 2. Materials and Methods 

### 2.1. Study Site

We conducted our study in Claremont, California, a suburb of Los Angeles situated at the eastern edge of Los Angeles County. The city is primarily residential, with interstitial parks, gardens, and other open spaces. The local climate is Mediterranean, with mild winters (with an average minimum temperature of 4 °C), during which almost all the annual precipitation occurs (with an average annual rainfall of 45.5 cm), followed by hot, dry summers (with an average maximum temperature of 32 °C). Despite the long summer drought, Claremont has been designated a Tree City by the National Arbor Day Association for 22 consecutive years, thanks in large part to widespread irrigation and horticulture on public and private properties. 

We centered our arthropod sampling around the Robert J. Bernard Biological Field Station (BFS), a 35 ha research station belonging to the Claremont University Consortium. California sage scrub, composed primarily of drought-tolerant shrubs (e.g., *Artemisia californica*, *Eriodictyon trichocalyx*, *Eriogonum fasciculatum*), covers 25 ha of the BFS. An additional 3.5 ha are non-native grassland dominated by *Bromus* spp. The last 5.5 ha were burned in a wildfire after sampling began. As a result, we did not include data from that area in our analyses. The fire did not impact BFS ground-dwelling arthropod communities [6,21]. 

We also sampled sites in the suburbs surrounding the BFS where humans directly and actively modified the landscape. To try to capture as much suburban heterogeneity as possible, we selected a diverse array of suburban locations within 30 to 1000 m of the BFS, including private residences, the Claremont Colleges, the Rancho Santa Ana Botanic Garden, and a small (~ 1 ha) administrative site at the edge of the BFS. In most cases, these sites were dominated by non-native plant species, with the notable exception of the Rancho Santa Ana Botanic Garden, which harbors only California-native plants (although its managed gardens mimic ecosystems from across the state, and therefore include numerous species not native to the Los Angeles County). In every case, the suburban sites received regular water subsidies. 

### 2.2. Sampling

We sampled terrestrial arthropod assemblages using pitfall traps at 16 sage-scrub, 8 grassland, and 16 suburban sites (see [18]). At each site, we placed three traps (3.2 cm in diameter, 25 cm deep) in an equilateral triangle with 10 m sides, except at six suburban sites, where obstacles (buildings, roads, sidewalks, etc.) prevented it. We sampled during five two-week periods in 2013 and 2014 roughly corresponding with the seasons (spring 2013: 29–30 March to 11–12 April; summer 2013: 1–3 to 15–17 July; fall 2013: 28 September to 12 October; winter 2014: 14–16 to 28–30 January; and spring 2014: 12–14 to 26–28 March). We identified all arthropods to the lowest taxonomic level possible. For taxa for which good taxonomic information exists (e.g., ants, isopods, earwigs), most (>95%) individuals were identified to the species level. For other arthropod groups (e.g., beetles) many individuals were only identified to the family level and placed into morpho-species groups. 

### 2.3. Analyses

It can be difficult to determine a species’ native range, particularly for arthropods [6,22]. For example, some species we collected are considered native to the state of California, but are unlikely to be native to Claremont (e.g., tree-dwelling *Camponotus* species). As such, we took a conservative approach and only included species with well-documented native ranges (main sources include [23,24]) in our analyses. This conservative approach means that some non-natives were probably excluded from our analyses but ensures that no native species were included.

To visualize the distribution of non-native species, we graphed the mean proportion of sites occupied across the five sampling periods for each habitat. We did the same for the mean number of individuals per pitfall trap. To test whether non-native species were consistently more widespread in a particular habitat, we ran a one-factor univariate PERMANOVA test using the average proportion of sites occupied in each habitat each season. Each PERMANOVA test used a resemblance matrix constructed using Euclidian distances in the program PRIMER_E v.6 with the PERMANOVA+ add-on [25]. We only ran PERMANOVA tests on species that were collected in at least 18 sites over the 5 seasons. We performed pair-wise comparisons using permutation-based T-tests following significant PERMANOVA tests. To test whether non-native species were more widespread in urban areas, we ran sign tests, scoring each species as a “+” if they were more widespread in suburban areas, or a “-” if they were more widespread in either the sage scrub or grassland habitat. We took two approaches to scoring a species as either a “+” or a “-”. The more conservative approach assigned species based on the outcome of the PERMANOVA tests and pair-wise comparisons, and therefore only included the species that were found in at least 18 sites. The more liberal approach scored species based on the visual assessment of graphs (i.e., whether a species appear to be more widespread in the suburban habitat than in either habitat in the BFS) and included species collected at fewer than 18 sites. The patterns in average abundance were similar to the patterns in the proportion of sites occupied and were intended only to supplement those data, so we did not run any statistical tests on those data.

## 3. Results

Out of 253 arthropod species collected, we identified 20 (~ 8 %) that were definitively non-native (Figure 1). Nearly half were insects (9 species), with arachnids (6 species), isopods (3 species), and myriapods (2 species) comprising the rest. Eight (40 % of the non-natives: *P. dilatatus, O. gracilis*, *M. simoni*, *T. barbatus*, *B. orientalis*, *C. mauritanica*, *E. annulipes*, *M. subulicola*) were collected exclusively in the suburban habitat. One (*C. septempunctata*) was only collected in the non-native grassland. 

Six of the eight non-native species found at ≥18 sites occupied a greater fraction of suburban sites than BFS sites in either habitat (Figure 1). The other two widespread species were not differently distributed among habitats. Sign tests (conservative approach, p = 0.031; liberal approach, p < 0.001) showed that non-native species were more widespread in the suburban habitat than in habitats at the BFS. Patterns in abundance were similar to patterns in site occupancy. Sixteen of the 20 non-native species had higher mean abundances in suburban pitfall traps than in traps in either habitat type at the BFS (Appendix A).

## 4. Discussion

Non-native species representing multiple arthropod taxa were less widespread and abundant in patches of sage scrub and non-native grassland than the suburban habitat surrounding them. Given the proximity of the sampling sites and lack of barriers between them, inadequate propagule pressure into the BFS is an unlikely explanation for this discrepancy. Recently established or transitory populations of non-natives in the suburban habitat are also improbable causes. Many of the non-natives we collected were first recorded in the region many decades ago (e.g., *Euborellia annulipes* established in 1880 [24]) and were abundant season after season at the same sites, suggesting that their populations are widespread and stable (i.e., they are not ephemeral populations associated with consistent propagule pressure). Instead, our findings suggest that characteristics of the habitats in the BFS enable them to resist invasion by non-native arthropods in the surrounding suburban habitat.

Abiotic factors are the most probable drivers of ecological resistance at the BFS. The sage-scrub and non-native grassland habitats differ greatly in their plant compositions, native arthropod assemblages [18], and other biotic variables (e.g., vertebrate assemblages; WM Meyer unpublished), but are both subject to the region’s hot, dry climate. Studies of plants [2,19,20] and Argentine ants [13,26] have highlighted the role of thermal and water stress in limiting invasions. We could not find studies of the desiccation tolerances of most of the non-native insects we collected, but our findings suggest these stressors likely increase the resistance of arid habitats to a wide range of arthropod invaders, particularly those that thrive in agricultural or urban environments where water subsidies are abundant. For example, *Armadillidium vulgare*, an isopod with particularly poor desiccation resistance [27], was the most widespread and abundant non-native species in the suburban habitat but was very rarely collected within the BFS, and then only in the spring, when cooler, wetter conditions prevailed. Likewise, vulnerability to desiccation and thermal stress may be responsible for the limited distribution of the wall spider, *Oecobius navus*, in the sage-scrub and grassland habitats [28]. Aridity may also indirectly increase resistance to invasion. The almost complete exclusion of isopods from the BFS likely limits the abundance of *Dysdera crocata*, the woodlouse spider, which frequently preys on them [29].

Because coastal habitat fragments in the region have more moderate climates, they may be more impacted by invasive arthropods. Previous studies have found that Argentine ants penetrate deeper into less arid coastal fragments and achieve high densities, especially within 100 m of urban boundaries [13,15]. They are comparatively scarce in drier inland fragments [13], including the BFS, where they were absent from more than half of the sampling sites each season, despite all sites being within 100 m of a developed edge [18]. Perhaps as a result of reduced competition with Argentine ants [30], arid fragments support higher native ant richness [13,18]. Complete or partial exclusion of Argentine ants may also benefit native non-ant arthropods [16,31], although the extent of their impact on non-ant taxa is unclear [32]. In addition to Argentine ants, our findings suggest that coastal fragments may also be more vulnerable to other arthropod invaders. Bolger et al. [16] found that the two most abundant non-ant arthropods in 40 coastal sage scrub fragments in San Diego County were the non-native isopods *A. vulgare* and *Porcellio laevis*, which made up approximately 30% of all non-ant individuals they captured. In our study, non-native isopods accounted for more than half of the total non-ant capture in the suburban habitat but represented just 0.3% of individuals from the BFS.

## 5. Conclusions

Understanding the role of desiccation stress in limiting invasions of insects and other arthropods provides a framework for understanding and modeling their distributions across the region, and likely in other semi-arid and arid urban centers. Previous research has shown that drier sage scrub and grassland habitat fragments resist invasion by Argentine ants better than more moderate fragments of the same size, and that reduced competition with Argentine ants results in greater native ant richness [13,18]. Our findings suggest that these arid fragments also resist invasion by many other non-native arthropods common in southern California suburbs. This may shelter native species from competition with and predation by non-natives, although such interactions are poorly understood. If so, inland habitats may be of high conservation value for arthropods. Decreasing water subsidies in open spaces around residential and commercial developments may also help mitigate the spread and impacts of arthropod invaders. However, to be effective, decreases in water subsidies would likely need to be significant. For example, both *A. vulgare* and *L. humile* were highly abundant in ‘water-wise’ gardens that used mulch to retain soil moisture, indicating that this and other water retention methods create conditions favorable enough to support these species. If adopted on a regional scale, a landscaping paradigm featuring drought-tolerant native plants that require little or no water subsidies might prevent the establishment of new non-native species and limit the abundance and spread of existing ones.

## Figures and Tables

**Figure 1 insects-10-00029-f001:**
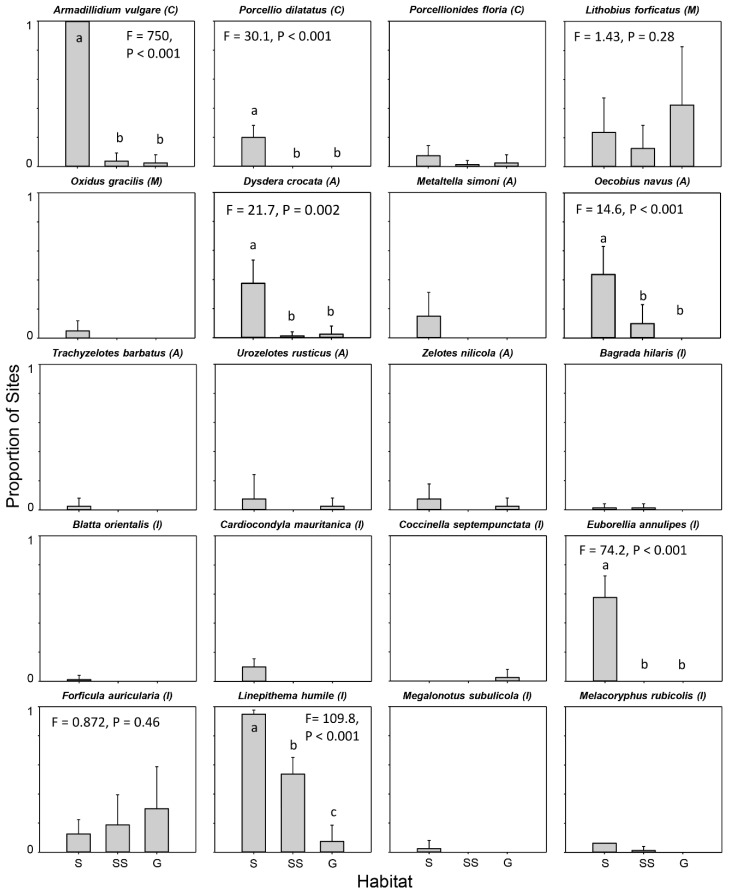
Site occupancy for each non-native species. Bar height is the mean proportion of sites occupied in each habitat type (S = Suburban; SS = Sage scrub; G = Grassland) over the five sampling periods. Error bars show standard deviation. Pseudo-F statistics and P-values from PERMANOVA tests are reported for those species that were collected in at least 18 sites over the 5 sampling periods. Letters on or above bars indicate significantly different means based on pair-wise comparisons. Letters next to species names indicate subphylum or class: C = Crustacea; M = Myriapoda; A = Arachnida; I = Insecta.

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
