# Peer review of "Distributions of Invasive Arthropods across Heterogeneous Urban Landscapes in Southern California: Aridity as a Key Component of Ecological Resistance"

_insects, 2019, doi:10.3390/insects10010029_

Round 1

Reviewer 1 Report

This is a rather simple "story" but clearly told. Under S Californian conditions, drought stress (aridity) acts as a powerful environmental filter against invasions by non-native arthropods. The study design is acceptable, the evaluaiton is not very sophisticated and uses simple statistical tests, but this is still acceptable.

A few minor comments are on the MS and below:

79-85 It'd be good to have some more quantiative dat on the plant coverage. As it is, this is only a qualitative declaration, a bit superficial and binary ("area was previously planted with non-natives") - more precise data are needed. Also some data on the amount of water irrigated would be needed.

89-90 how many of these sampling triangles had to be distorted or had to remain imperfect?

93 this is rather lax. How many of them were possible to identify to species (which families ro orders?) and how many to others? What was the crudest resolution, and how many grops/individuals had to be kept at this level?

Fig 1: no need to include species that were captured at only one habitat type. Revise the figure and delete D.crocata, O.gracilis, M.simoni, T.barbatus, B.orientalis, C.mauritanica, C.septempunctata, M.subulicola - these should be mentioned in the text only.

Author Response

Review 1:

This is a rather simple "story" but clearly told. Under S Californian conditions, drought stress (aridity) acts as a powerful environmental filter against invasions by non-native arthropods. The study design is acceptable, the evaluaiton is not very sophisticated and uses simple statistical tests, but this is still acceptable.

A few minor comments are on the MS and below:

79-85 It'd be good to have some more quantiative dat on the plant coverage. As it is, this is only a qualitative declaration, a bit superficial and binary ("area was previously planted with non-natives") - more precise data are needed. Also some data on the amount of water irrigated would be needed.

We understand your desire to know more about our suburban study sites. However, this information has been omitted because (1) it would be difficult to briefly and accurately describe the diverse array of suburban plant assemblages, and (2) a detailed description of those assemblages would add little to the story and distract from the primary focus. In this study, it is not a common plant assemblage or level of coverage that unite the suburban sites. Instead, it is significant human modification of the environment, notably via regular water subsidies. Quantitative data on the amount of water subsidies might be more useful, but those measurements were not taken. Arguably, such data would likely not provide any new insights, as it appears that even low levels of irrigation are enough to support common non-native arthropods (see lines 199-205). We agree that there is value to understanding how plant assemblages and thresholds of water use may influence the distributions of non-native species, and we hope that future studies assess those factors. But the data presented here are not fine enough to address the influence of specific plant assemblages and soil moisture levels.

89-90 how many of these sampling triangles had to be distorted or had to remain imperfect?

Six of the sixteen were not in the form of a triangle. We added this to the manuscript.

93 this is rather lax. How many of them were possible to identify to species (which families ro orders?) and how many to others? What was the crudest resolution, and how many grops/individuals had to be kept at this level?

We added wording to provide clarity.

Fig 1: no need to include species that were captured at only one habitat type. Revise the figure and delete D.crocata, O.gracilis, M.simoni, T.barbatus, B.orientalis, C.mauritanica, C.septempunctata, M.subulicola - these should be mentioned in the text only.

We respectfully disagree. We think that visual assessment of graphs is the most straightforward way to understand the distributions of non-native arthropods, even those that were only present in a single habitat type. To meet you half way, we added these species into the results section, indicating that they were only collected in the suburban site.

Reviewer 2 Report

I have reviewed MS “insects-416358-peer-review-v1”

The paper is very well-written, and needs little, I believe, in the way of grammatical changes. Argument for the research is sound and well-justified. I do, however, have criticisms about how the data are presented, and especially how they are analyzed.

Data were analyzed by Kruskal-Wallis (KW). This is a non-parametric procedure equivalent to it’s parametric equivalent, the 1-way analysis of variance (ANOVA). Non parametric procedures are performed when an author believes data violate one or more of the assumptions of a dataset and that the data cannot be analyzed with a parametric procedure. The most common violation is the dataset’s distribution (non normality/bell shaped curve). That is fine, but non-parametric procedures analyze differences among the MEDIANS of two or more sets of numbers. A parametric procedure, such as an ANVOA, analyzes the MEANS among two or more set of numbers. The two cannot be mixed. For this reason, Figure 1 is inappropriate. The authors report MEAN proportions, but the data were analyzed by a non-parametric procedure. This is inappropriate. Means should NOT be reported. And, what was analyzed. Numbers of critters trapped, or proportions?

If the point of the paper is to provide evidence that establishment of invasive species is more common in suburban areas, implicitly because of water, why split out each of the 20 species of invasives trapped into 20 different graphs and analyze them separately? A more powerful test would be to perform the KW, or maybe even an ANVOA, but to conduct the analysis not on proportions, but on the numbers trapped OVER all species and seasons combined. This would result in one bar graph (with 3 bars – S, SS, G) of numbers, say per trap. In other words, combine the numbers trapped but don’t split them by species. Compilation of data over all species and all five sampling dates, and analyzing those trap numbers (maybe even with an ANOVA) I think would be more powerful. As is, I don’t think the current Figure 1 adds much and may even detract from the MS.

Author Response

Review 2: our responses italicized

I have reviewed MS “insects-416358-peer-review-v1”

The paper is very well-written, and needs little, I believe, in the way of grammatical changes. Argument for the research is sound and well-justified. I do, however, have criticisms about how the data are presented, and especially how they are analyzed.

Thank you. We worked hard to elucidate the patterns observed as we think they are of significant ecological importance. We want this manuscript to inspire future research examining impacts of these habitat modifications. We address you concerns with our statistical approach below.

Data were analyzed by Kruskal-Wallis (KW). This is a non-parametric procedure equivalent to it’s parametric equivalent, the 1-way analysis of variance (ANOVA). Non parametric procedures are performed when an author believes data violate one or more of the assumptions of a dataset and that the data cannot be analyzed with a parametric procedure. The most common violation is the dataset’s distribution (non normality/bell shaped curve). That is fine, but non-parametric procedures analyze differences among the MEDIANS of two or more sets of numbers. A parametric procedure, such as an ANVOA, analyzes the MEANS among two or more set of numbers. The two cannot be mixed. For this reason, Figure 1 is inappropriate. The authors report MEAN proportions, but the data were analyzed by a non-parametric procedure. This is inappropriate. Means should NOT be reported. And, what was analyzed. Numbers of critters trapped, or proportions?

While we feel that the KW tests are appropriate, to alleviate concerns raised by this reviewer, we re-analyzed the data using PERMANOVA, a permutation-based alternative to a 1-Factor ANOVA, which specifically tests for differences among and between categorical groups and does not use rank data. We think it is appropriate to report means in the figure and use this test to indicate that some species are statistically more widespread in particular habitat types. The PERMANOVA tests produce nearly identical results as the original KW tests, suggesting that the patterns presented here are robust.

If the point of the paper is to provide evidence that establishment of invasive species is more common in suburban areas, implicitly because of water, why split out each of the 20 species of invasives trapped into 20 different graphs and analyze them separately? A more powerful test would be to perform the KW, or maybe even an ANVOA, but to conduct the analysis not on proportions, but on the numbers trapped OVER all species and seasons combined. This would result in one bar graph (with 3 bars – S, SS, G) of numbers, say per trap. In other words, combine the numbers trapped but don’t split them by species. Compilation of data over all species and all five sampling dates, and analyzing those trap numbers (maybe even with an ANOVA) I think would be more powerful. As is, I don’t think the current Figure 1 adds much and may even detract from the MS.

We have a different opinion. To our understanding, the reviewer believes the “point of the paper is to provide evidence that establishment of invasive species is more common in suburban areas.” However, it is our opinion that the goal of the paper is to determine whether there are consistent patterns in the distributions among the three habitat types of a wide variety of evolutionarily distinct non-native arthropods. We agree that performing a KW test or ANOVA on the total number of individuals in each habitat type across all seasons and species would be a more statistically powerful test. However, this would come at the cost of valuable information about how representative the overall trend is of the distributions of individual invasive species, and could result in the distributions of the most abundant species (e.g., L. humile and A. vulgare) masking significant contrary patterns in the distributions of less numerous species.

Moreover, we feel that abundance, abundance per trap, or abundance per site are not the most useful metrics of non-native species distributions. Especially for colonial species, such as ants, abundance per trap may be more related to the distance of a trap to a nest or colony than the actual abundance of that species. This is partly why we designed our sites to include three traps in close proximity to each other, rather than a single trap. We think the proportion of traps occupied at a site or proportion of sites occupied in a habitat are better measures of how a species is distributed. Still, we wholeheartedly agree that some measure of the number of individuals captured is an important part of the picture, which is why we included graphs of average abundance per trap in each habitat type for each species as a supplemental figure.

For clarity, we did run a global test (the sign test) that specifically examines if this pattern of non-native species being more widespread in suburban areas is consistent across multiple species and may be predictive of future invasions. The individual KW, now PERMANOVA tests, were run for two reasons: (1) to examine which species were more widespread in a particular habitat, and (2) inform our conservative sign test. As such, we strongly believe that it is more conservative and ecologically relevant to examine the 20 species individually, then examine if patterns are consistent across species.